materials science/energy

composite polymer electrolyte, *in situ*, matrix regulating, all-solid-state, battery

**Authors for correspondence:**
Yang Liu
e-mail: liuyang81@shu.edu.cn
Jingjing Zhou
e-mail: zjajzjj@t.shu.edu.cn

[†]These authors contributed equally to this paper.

This article has been edited by the Royal Society of Chemistry, including the commissioning, peer review process and editorial aspects up to the point of acceptance.

# A polycarboxylic/ether composite polymer electrolyte via *in situ* UV-curing for all-solid-state lithium battery

Wenli Wang[1,†], Ziwen Qiu[1,†], Qian Wang[1], Xiaoyu Zhou[1], Yang Liu[1,2], Jingjing Zhou[1] and Bingkun Guo[1]

[1]Materials Genome Institute, Shanghai University, Shanghai 200444, People's Republic of China
[2]Guangdong Provincial Key Laboratory of Energy Materials for Electric Power, Southern University of Science and Technology, Shenzhen 518055, People's Republic of China

YL, 0000-0001-5738-0049

A polycarboxylic/ether composite polymer electrolyte derived from two-arm monomer and polyethylene oxide (PEO) was *in situ* synthesized on the cathode. The composite electrolyte exhibits a high ionic conductivity of $3.6 \times 10^{-5}$ S cm$^{-1}$, high oxidation stability, excellent stability towards Li metal and makes Li/LiFePO$_4$ present good cyclic and rate performance at 25°C.

## 1. Introduction

The development of portable electronics and electric vehicles requires small/tiny form factor energy storage devices with high energy density and stable operability [1,2]. Thus, solid-state batteries (SSBs) have been considered as one of the most promising technologies for next-generation energy storage devices with high energy density and high safety [3]. The solid-state electrolytes (SSEs) are the key component for SSBs, which generally possess various advantages compared with conventional liquid electrolytes, including nonflammable, non-volatile, no leakage risk and high compatibility with metallic lithium [4].

SSEs can be briefly divided into two categories: inorganic ion conductors such as oxides and sulfides, and polymers such as polyesters, polyethers, polyolefins and their derivatives [5,6]. Each of them displays distinctive advantages and disadvantages. The oxides show high chemical/electrochemical stability and relatively high ionic conductivity (greater than

**Figure 1.** FTIR (*a*) and DSC (*b*) curves of the SSE films.

$10^{-4}$ S cm$^{-1}$), while they suffer from large interfacial resistance and brittleness [7,8]. Sulfides have very high ionic conductivity (greater than 25 mS cm$^{-1}$) [9,10], but they are thermodynamically unstable with lithium metal and sensitive to the air [11]. By contrast, the polymer electrolytes possess high flexibility, high chemical stability and good compatibility with electrodes, revealing great application potential, especially polyethylene oxide (PEO)-based polymer electrolytes [12]. However, polymer electrolytes usually present low intrinsic ionic conductivity at room temperature (RT) and high interfacial impedance due to poor contact between electrolytes and electrodes [13,14]. Many strategies have been proposed to increase the conductivity of polymer electrolytes and great progress has been made [15,16]. Sun and co-workers [17] reported the poly(ethylene glycol)–poly(etheramine)-based interpenetrating network polymer electrolyte. The electrolyte shows an ionic conductivity of $5.6 \times 10^{-5}$ and $1.1 \times 10^{-3}$ S cm$^{-1}$ at 25 and 100°C, and makes the LiFePO$_4$ lithium metal batteries present an initial discharge capacity of 156.2 mA h g$^{-1}$ and stable cycling performance over 200 cycles at 0.1 C. Li and co-workers [18]. reported a PEO-based solid-state electrolyte composited with polydopamine (PDA)-coated Li$_{6.4}$La$_3$Zr$_{1.4}$Ta$_{0.6}$O$_{12}$ nanoparticles The SSE shows a conductivity of $1.1 \times 10^{-4}$ S cm$^{-1}$ at 30°C and good compatibility adhesion with both positive and negative electrodes.

*In situ* polymerization is considered to be a facile and effective approach to improve the contact between electrolytes and electrodes [19–21]. In our previous work [22], we reported an *in situ* method to construct composite electronlyes on the cathodes by UV-curing, which effectively reduced the interfacial impedance by 69.1%. Benefiting from that, the all-solid-state LiFePO$_4$/Li cell displayed a good electrochemial performance at RT. However, the ionic conductivity of the composite electrolytes are still low (approx. $2.21 \times 10^{-5}$ S cm$^{-1}$ at 25°C), resulting in a poor rate property. Here, we used di(ethylene glycol) diacrylate (A2) as monomer to generate composite electrolytes with PEO via UV-curing. The composite electrolytes built from A2 exhibited a high conductivity of $3.6 \times 10^{-5}$ S cm$^{-1}$ at RT, that is 60% higher than that of the composite electrolytes derived from three-arm or four-arm methyl acrylate monomer [22]. The composite electrolytes also show good dendrite suppression property and large electrochemical window. Moreover, The all-solid-state LiFePO$_4$/Li cells with the composite electrolyte display a high capacity of approximately 155 mA h g$^{-1}$ and good capacity retention at RT.

## 2. Results and discussion

Fourier transform infrared spectroscopy (FTIR) was used to investigate the polymerization of di(ethylene glycol) diacrylate (A2) and the interaction between poly(di(ethylene glycol) diacrylate) (PA2) and PEO. As shown in figure 1*a*, the peaks between 1610 and 1640 cm$^{-1}$ in the spectrum of monomer A2 are related to the C=C groups, and no peak can be detected in the same spectrum region of PA2 and PA2 with 30 wt% PEO (PA2-30%PEO). Combined with the literature [22,23], these can be understood as the polymerization of acrylate groups via UV-curing. The peaks at approximately 1050 cm$^{-1}$ can be attributed to the C-O group of PEO. The C-O peaks of PEO and PA2 are all shifted in the FTIR spectrum of PA2-30%PEO, suggesting the interaction between the polymer matrix PA2 and PEO. Compared with the C-O peak of PEO, it is obvious the similar peak of PA2-30%PEO shifts to lower wavenumber region. This should be related to the electronic levelling effect

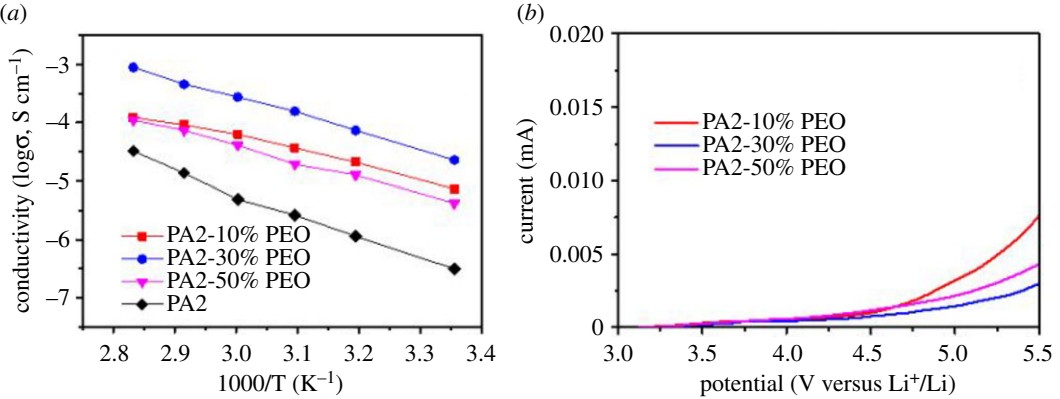

**Figure 2.** Ionic conductivities (*a*) and LSV curves (*b*, scan rate: 0.5 mV s$^{-1}$) of the SSE films with different content of PEO.

cause by the interaction between PA2 and PEO, which reduces the energy of the carboxyl group in PA2-30%PEO. The differential scanning calorimetry (DSC) patterns of the samples are presented in figure 1*b*. The glass transition temperature (Tg) of PA2-30%PEO is higher than that of PEO, suggesting PA2-30% PEO is more likely to present higher disorder degree at RT.

Ionic conductivities of the samples with different content of PEO were measured by AC impedance technique as shown in figure 2*a*. PA2-30%PEO shows the highest ionic conductivity of $3.6 \times 10^{-5}$ S cm$^{-1}$ at 25°C, which is 60% higher than that of previous report [22]. The conductivities of composite electrolytes obviously decrease when the content of PEO is further increased to 50 wt% or decreased to 10 wt%. The electrochemical windows of the SSEs were tested with the stainless steel (SS)/SSE/Li cells by linear sweep voltammetry (LSV; figure 2*b*) at RT. The LSV results show the similar trend as that of conductivity. The composite with 30 wt.% PEO added presents the highest oxidation potential without any obvious oxidation peak until approximately 4.7 V versus Li$^+$/Li. The sample also exhibits the smallest oxidation current while charged at the potential higher than 5.0 V versus Li$^+$/Li.

Then X-ray diffraction (XRD) was used to study the influence of PEO content on the crystallinities of composites. As shown in electronic supplementary material, figure S1, it is clear that PA2-30%PEO still shows amorphous property compared with PA2 and PA2-10%PEO, suggesting the homogeneous distribution of each component in PA2-30%PEO. When the content of PEO is up to 50 wt%, sharp peaks which should be ascribed to PEO can be observed in the pattern of PA2-50%PEO, meaning the higher crystallinity of the sample [22]. Generally, the lower crystallinity provides the larger wriggling space for polymer chains, which is benefit for improving the ionic conductivity of SSE. Compared with other samples in electronic supplementary material, figure S1, the relative low crystallinity of PA2-30%PEO also means the excellent interaction between PA2 and PEO without components segregation. The interaction results in the electronic levelling effect, which makes the widest electrochemical window of PA2-30%PEO among the samples. The low ionic conductivity of PA2-10% PEO should be related to the poor ionic conductivity of PA2 at RT (figure 2*a*), even if it possesses an amorphous property.

The surface morphology of PA2-30%PEO *in situ* synthesized on LiFePO$_4$ cathode was examined by the scanning electron microscope (SEM). PA2-30%PEO presents a smooth surface and no aggregation can be observed, as shown in figure 3*a*, meaning the homodisperse of the composite electrolyte. The compatibility of PA2-30%PEO with lithium metal anode was investigated by Li/SSE/Li cell in figure 3*b* at the current density of 0.10 mA cm$^{-2}$. This cell presents the polarization potential of approximately 0.25 V without a significant increase in the cycling more than 230 h, suggesting the good electrochemical and thermodynamics stabilities between PA2-30%PEO and lithium metal.

Li/PA2-30%PEO/LiFePO$_4$ cells were assembled and tested at RT to further study the composite electrolyte's electrochemical properties while the SSEs were *in situ* constructed on the cathodes by UV-curing. As shown in figure 4*a*, the cell exhibits a high reversible capacity of approximately 155 mA h g$^{-1}$ with the initial coulombic efficiency of 94.92% at 0.1 C. The capacity retention of this cell is approximately 86% in 50 cycles with the coulombic efficiency of approximately 100%, indicating the good cycling stability of the composite electrolyte. Furthermore, the all-solid-state lithium ion battery also presents the good rate performance at RT. It exhibits the initial discharge capacity of 156.2 mA h g$^{-1}$ at 0.05 C and keeps a remarkable discharge capacity of 142.5 mA h g$^{-1}$ at 0.2 C. At the rate of 0.5 C, the discharge capacity of the cell drops to 85 mA h g$^{-1}$, but still much

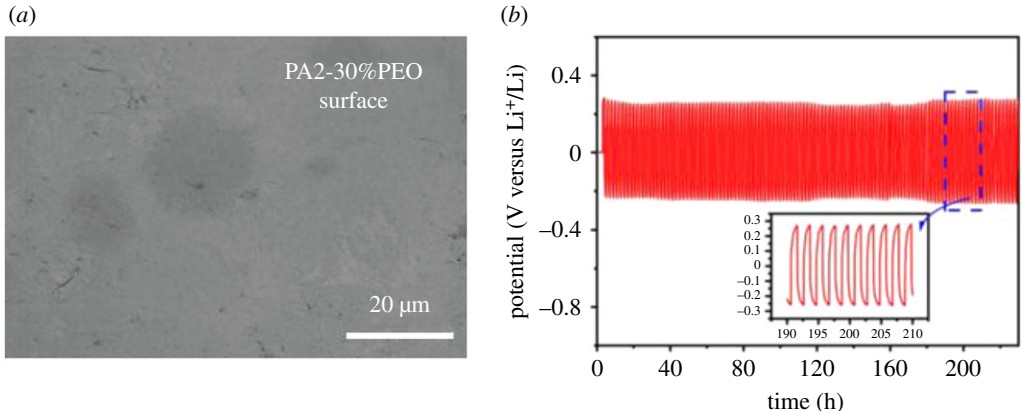

**Figure 3.** (a) The SEM image of PA2-30%PEO-coated electrode surface and (b) the galvanostatic cycling of the symmetrical cell at RT.

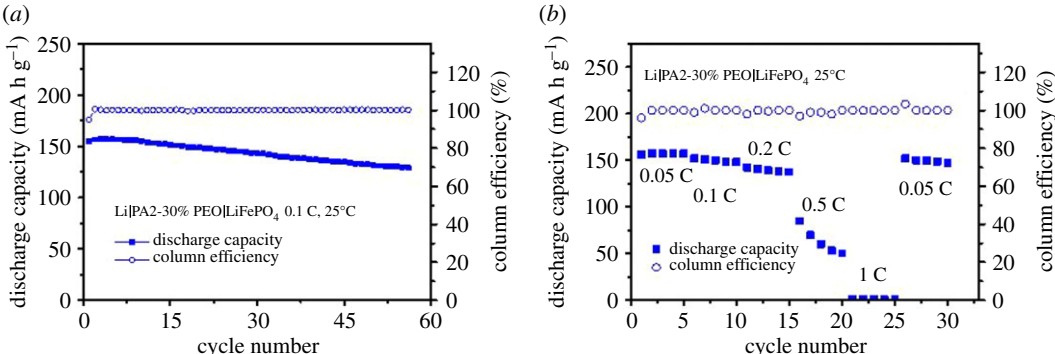

**Figure 4.** The cyclic and rate performances of Li/PA2-30%PEO/LiFePO$_4$ at RT.

higher than that of the cells with the composites formed from multiple-arm monomers as reported in our early work [22], meaning this idea, regulating the matrix structure by adjusting the monomer topology for enhancing the electrochemical properties of SSEs, is feasible and effective. When the rate returns to 0.05 C, the discharge capacity of the cell is almost all recovered.

# 3. Conclusion

In summary, a novel composite SSE has been developed by adjusting monomer topology and *in situ* synthesized via UV-curing on cathode. The SSE PA2-30%PEO presents the highest ionic conductivities of $3.6 \times 10^{-5}$ S cm$^{-1}$ at RT, which is 60% higher than the three-arm monomer derived composite electrolyte. The PA2-30%PEO composite electrolyte also exhibits excellent compatibility with Li electrode and good dendrite suppression property. The Li/PA2-30%PEO/LiFePO$_4$ cell shows the high capacity of 155 mA h g$^{-1}$ under 0.1 C at 25°C and long cycling stability. Meanwhile, the cell with PA2-30%PEO also displays good rate capability with remarkable improvement compared with the similar SSE we reported at RT [22]. Considering many works fail to reveal the electrochemical performance of solid cells at RT and only show the results at high temperatures (greater than or equal to 60°C) [22], the composite polymer electrolyte formed from two-arm monomer would be an alternative SSE for solid-state lithium-based batteries, and the strategy that regulating the matrix structure by adjusting the monomer topology for improving the electrochemical properties of composite electrolytes would provide the inspiration for designing other high-performance SSEs.

Data accessibility. This article has no additional data.

Authors' contributions. W.W. and Z.Q. synthesized the polymer films and made most of the electrochemistry measures and SEM tests. Q.W. conducted and analysed the FTIR experiments. X.Z. made the DSC measurement and compared the results. Y.L. and J.Z. developed the ideas, set up appropriate experiments, wrote and revised the manuscript. B.G. helped revise the manuscript. All authors have given approval to the final version of the manuscript.

Competing interests. We declare we have no competing interests.

Funding. This work was supported by the Science and Technology Commission of Shanghai Municipality (grant no. 18010500-300) and Guangdong Provincial Key Laboratory of Energy Materials for Electric Power (grant no. 2018B030322001).

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
