## [Reviewer comments · Royal Society Open Science]

Review History

RSOS-200598.R0 (Original submission)

Review form: Reviewer 1

Is the manuscript scientifically sound in its present form?

Yes

Are the interpretations and conclusions justified by the results?

Yes

Is the language acceptable?

Yes

Do you have any ethical concerns with this paper?

No

Have you any concerns about statistical analyses in this paper?

No

Recommendation?

Accept with minor revision (please list in comments)

Comments to the Author(s)

In this manuscript, a composite polymer electrolyte generates on the cathode via in situ UV-curing and the ionic conductivity is increased by regulating the matrix structure. The composite electrolyte displays good electrochemical properties in the solid-state cells. It can be published after addressing the following minor issues.

1. More discussion about the interaction between PA2 and PEO should be given.
2. The authors claim that the good ionic conductivity of PA2-30%PEO is related to the lower crystallinity (Figure S1). However, the PA2-10%PEO, possessing an amorphous property as shown in Figure S1, displays a low conductivity. The authors need to give more explanation.
3. The scan rate of LSV should be added.

Review form: Reviewer 2**Is the manuscript scientifically sound in its present form?**

No

Are the interpretations and conclusions justified by the results?

No

Is the language acceptable?

Yes

Do you have any ethical concerns with this paper?

No

Have you any concerns about statistical analyses in this paper?

No

Recommendation?

Reject

Comments to the Author(s)

Novelty of electrolyte material is not very clear, as it is so common class of materials. Room temperature ionic conductivity is very low to expect good cell performance. In fact, moderate discharging capacity was observed only when charge-discharge rate was very low, and observed value at 1C was extremely low. This reviewer does not consider this results will be appreciated by the community.

Decision letter (RSOS-200598.R0)

Dear Dr Liu:

Title: A polycarboxylic/ether composite polymer electrolyte via in situ UV-curing for all-solid-state lithium battery
Manuscript ID: RSOS-200598

The editor assigned to your manuscript has now received comments from reviewers. We would like you to revise your paper in accordance with the referee and Subject Editor suggestions which can be found below (not including confidential reports to the Editor). Please note this decision does not guarantee eventual acceptance.

Please submit your revised paper before 19-Jun-2020. Please note that the revision deadline will expire at 00.00am on this date. If we do not hear from you within this time then it will be assumed that the paper has been withdrawn. In exceptional circumstances, extensions may be possible if agreed with the Editorial Office in advance. We do not allow multiple rounds of revision so we urge you to make every effort to fully address all of the comments at this stage. If deemed necessary by the Editors, your manuscript will be sent back to one or more of the original reviewers for assessment. If the original reviewers are not available we may invite new reviewers.

On behalf of the Subject Editor Professor Anthony Stace and the Associate Editor Dr Chaohua Cui.

RSC Associate Editor:

Comments to the Author:

I am concerned about the comment from the reviewer #2: "moderate discharging capacity was observed only when charge-discharge rate was very low". Please make a clear explanation regarding this point.

RSC Subject Editor:

Comments to the Author:

(There are no comments.)

Reviewers' Comments to Author:

Reviewer: 1

Comments to the Author(s)

In this manuscript, a composite polymer electrolyte generates on the cathode via in situ UV-curing and the ionic conductivity is increased by regulating the matrix structure. The composite electrolyte displays good electrochemical properties in the solid-state cells. It can be published after addressing the following minor issues.

1. More discussion about the interaction between PA2 and PEO should be given.
2. The authors claim that the good ionic conductivity of PA2-30%PEO is related to the lower crystallinity (Figure S1). However, the PA2-10%PEO, possessing an amorphous property as shown in Figure S1, displays a low conductivity. The authors need to give more explanation.
3. The scan rate of LSV should be added.

Reviewer: 2

Comments to the Author(s)

Novelty of electrolyte material is not very clear, as it is so common class of materials. Room temperature ionic conductivity is very low to expect good cell performance. In fact, moderate discharging capacity was observed only when charge-discharge rate was very low, and observed value at 1C was extremely low. This reviewer does not consider this results will be appreciated by the community.

Author's Response to Decision Letter for (RSOS-200598.R0)

See Appendix A.

Decision letter (RSOS-200598.R1)

Dear Dr Liu:

Title: A polycarboxylic/ether composite polymer electrolyte via in situ UV-curing for all-solid-state lithium battery
Manuscript ID: RSOS-200598.R1

It is a pleasure to accept your manuscript in its current form for publication in Royal Society Open Science. The chemistry content of Royal Society Open Science is published in collaboration with the Royal Society of Chemistry.

On behalf of the Subject Editor Professor Anthony Stace and the Associate Editor Dr Chaohua Cui.

RSC Associate Editor
Comments to the Author:
(There are no comments.)

Reviewer(s)' Comments to Author:

Appendix A

Reviewers' Comments to Author:

Reviewer: 1

Comments to the Author(s)

In this manuscript, a composite polymer electrolyte generates on the cathode via in situ UV-curing and the ionic conductivity is increased by regulating the matrix structure. The composite electrolyte displays good electrochemical properties in the solid-state cells. It can be published after addressing the following minor issues.

1. More discussion about the interaction between PA2 and PEO should be given.

Reply:

The C-O peaks of PEO and PA2 are all shifted in the FTIR spectrum of PA2-30%PEO, suggesting the interaction between the polymer matrix PA2 and PEO. Compared to the C-O peak of PEO, it is obvious the similar peak of PA2-30%PEO shifts to less wavenumber region. This should be related to the electronic leveling effect cause by the interaction between PA2 and PEO, which reduces the energy of the carboxyl group in PA2-30%PEO.

The discussion has been updated in the manuscript.

2. The authors claim that the good ionic conductivity of PA2-30%PEO is related to the lower crystallinity (Figure S1). However, the PA2-10%PEO, possessing an amorphous property as shown in Figure S1, displays a low conductivity. The authors need to give more explanation.

Reply:

The low ionic conductivity of PA2-10%PEO should be related to the poor ionic conductivity of PA2 at room temperature, even if which possesses an amorphous property.

The discussion has been updated.

3. The scan rate of LSV should be added.

Reply:

The scan rate is 0.5 mV s⁻¹ and has been updated in the manuscript.

Reviewer: 2

Comments to the Author(s)

Novelty of electrolyte material is not very clear, as it is so common class of materials. Room temperature ionic conductivity is very low to expect good cell performance. In fact, moderate discharging capacity was observed only when charge-discharge rate was very low, and observed value at 1C was extremely low. This reviewer does not consider this results will be appreciated by the community.

Reply:

In this work, a novel all-solid-state electrolyte (SSE) has been in situ

*synthesized on the surface of porous electrode via regulating the matrix structure. The SSE PA2-30%PEO presents the higher ionic conductivity of $3.6 * 10^{-5} \text{ S cm}^{-1}$ and rate performances than the reported work (ref22) at room temperature. Besides, the method of in situ polymerization via UC-curing effectively reduces the interfacial impedance of electrode/solid electrolyte, which ensures the solid cells workable at room temperature. By contrast, many works about solid polymer electrolytes fail to reveal the electrochemical performance of solid cells at room temperature and only show the results at high temperatures ($\geq 60 \text{ }^\circ\text{C}$) even if they present good ionic conductivities, such as the works shown in Table S2 of ref22.*

Our results exhibit the novelty of this work, and the strategy of enhancing ionic conductivity of SSEs by adjusting the monomer topology for enhancing the electrochemical properties can provide the inspiration for designing other high performance SSEs.

The discussion has been updated.